# Discovery, Pharmacological Characterisation and NMR Structure of the Novel µ-Conotoxin SxIIIC, a Potent and Irreversible Na_V_ Channel Inhibitor

**DOI:** 10.3390/biomedicines8100391

**Published:** 2020-10-02

**Authors:** Kirsten L. McMahon, Hue N.T. Tran, Jennifer R. Deuis, Richard J. Lewis, Irina Vetter, Christina I. Schroeder

**Affiliations:** 1Institute for Molecular Bioscience, The University of Queensland, Brisbane, QLD 4072, Australia; k.mcmahon@uq.edu.au (K.L.M.); hue.tran56@uq.net.au (H.N.T.T.); j.deuis@uq.edu.au (J.R.D.); r.lewis@uq.edu.au (R.J.L.); 2The School of Pharmacy, The University of Queensland, Woolloongabba, QLD 4102, Australia; 3National Cancer Institute, National Institutes of Health, Frederick, MD 21702, USA

**Keywords:** chronic pain, µ-conotoxin, Na_V_ channels, pore blocker, whole-cell patch-clamp electrophysiology, NMR

## Abstract

Voltage-gated sodium (Na_V_) channel subtypes, including Na_V_1.7, are promising targets for the treatment of neurological diseases, such as chronic pain. Cone snail-derived µ-conotoxins are small, potent Na_V_ channel inhibitors which represent potential drug leads. Of the 22 µ-conotoxins characterised so far, only a small number, including KIIIA and CnIIIC, have shown inhibition against human Na_V_1.7. We have recently identified a novel µ-conotoxin, SxIIIC, from *Conus striolatus*. Here we present the isolation of native peptide, chemical synthesis, characterisation of human Na_V_ channel activity by whole-cell patch-clamp electrophysiology and analysis of the NMR solution structure. SxIIIC displays a unique Na_V_ channel selectivity profile (1.4 > 1.3 > 1.1 ≈ 1.6 ≈ 1.7 > 1.2 >> 1.5 ≈ 1.8) when compared to other µ-conotoxins and represents one of the most potent human Na_V_1.7 putative pore blockers (IC_50_ 152.2 ± 21.8 nM) to date. NMR analysis reveals the structure of SxIIIC includes the characteristic α-helix seen in other µ-conotoxins. Future investigations into structure-activity relationships of SxIIIC are expected to provide insights into residues important for Na_V_ channel pore blocker selectivity and subsequently important for chronic pain drug development.

## 1. Introduction

Voltage-gated sodium (Na_V_) channels are highly conserved pore-forming proteins permeable to sodium ions and are important for the initiation and propagation of action potentials [1,2]. Nine different subtypes are expressed in humans (Na_V_1.1–1.9) and are encoded by the *SCNxA* genes (x = 1–5, 8–11) [3]. Genetic mutations in human Na_V_ channels give rise to a number of neurological disorders, for example, epilepsy (Na_V_1.1), myotonia (Na_V_1.4) and cardiac arrhythmia (Na_V_1.5) [4,5,6]. Moreover, patients with loss of function mutations in Na_V_1.7 present with congenital insensitivity to pain and aside from anosmia (loss of smell) have no serious physiological deficits, highlighting the potential of Na_V_1.7 as a drug target for pain therapeutics [7]. Despite compelling genetic evidence and studies implicating several additional subtypes, including Na_V_1.1, 1.3, 1.6, 1.8, and 1.9, as potential targets for pain therapies [8], the clinical utility of Na_V_ channel inhibitors has had limited success due to high sequence homology between Na_V_ channel subtypes, particularly within the pore domain [9], as off-target Na_V_ inhibition could lead to undesired side-effects. Therefore, a thorough understanding of subtype selectivity is essential for exploiting Na_V_ channel inhibitors as therapeutics. Toxins from venomous creatures, such as snakes, spiders, and cone snails, provide excellent pharmacological tools to study Na_V_ channels [10,11].

The venom of predatory marine cone snails represents a complex source of disulfide-rich bioactive peptides that modulate ion channels, called conotoxins [12]. One of the most numerous and best-characterised conotoxin classes are the µ-conotoxins, which have a distinctive type III cysteine framework (CC–C–C–CC). They are potent and selective Na_V_ channel inhibitors and µ-conotoxin research has led to many advances in the understanding of Na_V_ channel function. In 2001, Li et al. established the clockwise arrangement of the four domains of Na_V_ channels using μ-conotoxin GIIIA as a probe [13] and more recently, the structure of human Na_V_1.2 bound to μ-conotoxin KIIIA was solved by cryogenic electron microscopy (cryo-EM), confirming for the first time interactions between µ-conotoxins and the Na_V_ channel pore [14]. These recent developments have reinvigorated efforts to exploit µ-conotoxins as drug leads.

µ-Conotoxins represent favourable drug leads due to two key characteristics. First, they are rich in cysteines which form disulfide bonds to afford structural integrity and secondly, their small molecular size (typically 16–26 residues) presents an advantage over larger biologics as they are easily synthesised and amenable to chemical modifications to improve pharmacological traits [15]. Conversely, they present as favourable molecules over similarly acting small molecule Na_V_ inhibitors including anesthetics [16], as the increased surface area provides greater contacts with the Na_V_ channel pore, which can result in an increased subtype selectivity. To date, 22 µ-conotoxins have been described from 13 different *Conus* species [17]. µ-Conotoxins typically show a preference for Na_V_1.2 and Na_V_1.4 when evaluated in rat homologues, with the exception of BuIIIB, which shows a preference for Na_V_1.3 [18]. To date only a small number of µ-conotoxins have been found to inhibit human (h)Na_V_1.7, including KIIIA (IC_50_ 147 nm) [19] and CnIIIC (IC_50_ 485 ± 94 nM) [20]. One species receiving relatively little attention is the piscivorous (fish-hunting) *Conus striolatus.* Until recently only four peptide inhibitors (Sx11.2, Sx4.1, and µ-conotoxins SxIIIA, and SxIIIB) were reported from this species, with SxIIIA potently inhibiting Na_V_1.4 (IC_50_ 7 nM), but not Na_V_1.7 [21]. The activity of the remaining peptides has not been assessed to date.

In the current study, we present the discovery of a novel µ-conotoxin from *C. striolatus*, named SxIIIC. We have characterised its pharmacological activity as a Na_V_ channel inhibitor and evaluated the NMR solution structure. Our research discovered that SxIIIC displays a unique Na_V_ channel subtype selectivity profile when compared to other highly-homologous µ-conotoxins. Interestingly, at the therapeutically relevant subtype Na_V_1.7, SxIIIC displays nanomolar potency and near irreversible inhibition. To date, SxIIIC represents one of the most potent µ-conotoxin putative pore blocker inhibitors of hNa_V_1.7. Structural differences between SxIIIC and other µ-conotoxins may provide insight into residues driving potent Na_V_ channel inhibition. Future investigation of structure-activity relationships of these pore blockers may open avenues for the successful development of therapeutic Na_V_ channel inhibitors.

## 2. Experimental Section

### 2.1. Chemicals

All chemicals were purchased from Sigma Aldrich (Sigma Aldrich, St. Louis, MO, USA) and amino acids were purchased from Chem-Impex International (Chem-Impex International, Inc., Wood Dale, IL, USA) unless otherwise stated.

### 2.2. Activity-Guided Isolation of SxIIIC

Inhibition of Na_V_-mediated responses by crude and fractionated venom from *C. striolatus* was assessed using a FLIPR^TETRA^ (Molecular Devices, Sunnyvale, CA, USA) high-throughput assay, as previously described [22]. In brief, SH-SY5Y neuroblastoma cells were plated at a density of 120,000 cells/well on 96-well black-walled imaging plates (Corning Inc., Corning, NY, USA) in growth medium (RPMI medium: 15% fetal bovine serum and 1 mM L-glutamine) and cultured for 48 h. Cells were loaded with Calcium 4 No-wash dye (Molecular Devices) diluted in physiological salt solution (PSS; composition (in mM) 140 NaCl, 11.5 glucose, 5.9 KCl, 1.4 MgCl_2_, 1.2 NaH_2_PO_4_, 5 NaHCO_3_, 1.8 CaCl_2_, and 10 HEPES) by replacing growth medium with dye solution and incubating at 37 °C for 30 min. Fluorescence responses to the addition of crude venom or venom fractions were recorded every second for 300 s (excitation, 470–495 nm; emission, 515–575 nm) and responses to the addition of veratridine (50 µM) were recorded at 1 s intervals for a further 300 s.

Crude venom was isolated from one specimen of *C. striolatus* by stripping the venom duct contents. The crude venom was dissolved in 30% acetonitrile/0.1% formic acid, vortexed, and centrifuged at 10,000× *g* for 5 min to remove insoluble components. Crude venom (200 µg) was fractionated into 45 × 0.7 mL fractions (1 per min) using a Vydac, 5 µm C18 218TP, 250 × 4.6 mm column (Grace Davison Discovery Sciences, Columbia, MD, USA) eluted at a flow rate of 0.7 mL/min with 5–45% solvent B over 45 min (solvent A, water (H_2_O)/0.1% formic acid; solvent B, 90% acetonitrile/0.1% formic acid), with detection at 214 nm. Activity-guided fractionation was performed using the FLIPR^Tetra^ high-throughput Na_V_ assay described above. Further purification was carried out by size-exclusion chromatography (Superdex Peptide, HR 10/30; Amersham Biosciences, Little Chalfont, UK) using elution with 30% acetonitrile, 0.1% formic acid, aqueous at a flow rate of 0.5 mL/min, with detection at 214 nm. The resulting purified fraction was then analysed on a MALDI-TOF mass spectrometer (4700 Proteomics Analyzer; Applied Biosystems, Mulgrave, VIC, Australia) using α-cyano-4-hydroxycinnamic acid (5 mg/mL) as the matrix.

### 2.3. Peptide Sequencing

The purified peptide was sequenced using Edman degradation at the Australian Proteome Research Facility. Briefly, the peptide was dissolved in 4 M urea in 50 mM ammonium bicarbonate and reduced with 100 nM dithiothreitol at 56 °C for 1 h under argon. Subsequently, the sample was alkylated using 220 nM acrylamide for 0.5 h in the dark. The reaction was quenched by addition of excess dithiothreitol. Following desalting by reversed-phase high-performance liquid chromatography (RP-HPLC), the collected fraction was loaded onto pre-cycled bioprene discs and subjected to 35 cycles of Edman N-terminal sequencing using an Applied Biosystems 494 Procise Protein Sequencing System.

### 2.4. Peptide Synthesis

Synthetic SxIIIC (RGCCNGRGGCSSRWCRDHARCC*) was assembled by solid-phase peptide synthesis on a Liberty Prime automatic synthesiser (CEM, Matthews, NC, USA), on rink amide-AM resin (0.139 mmol/g) at a 0.1 mmol scale to produce C-terminal amidated peptide (*) using 9-fluorenylmethoxycarbonyl (Fmoc) methodology. Side-chain protecting groups used were Arg(Pbf), Asn(Trt), Cys(Trt), His(Trt), Ser(tBu), and Trp(Boc). The peptide was cleaved from resin and side chains simultaneously removed following a 2 h incubation with 92.5% trifluoroacetic acid (TFA)/2.5% anisole/2.5% triisopropylsilane (TIPS)/2.5% H_2_O. Excess TFA was evaporated by N_2_ flow and reaction precipitated with ice-cold diethyl ether. Precipitant was collected via gravity filtration on a 20 µM frit (Phenomenix, Torrance, CA, USA) and crude peptide dissolved in 50% solvent B and lyophilised. Linear SxIIIC was dissolved in <2% solvent B and loaded onto a Gemini, 5 µm C18 110 Å, 250 × 21.2 mm column (Phenomenex, Torrance, CA, USA). Following column equilibration (0–5% solvent B over 5 min), synthetic SxIIIC was purified by RP-HPLC using a linear gradient between 5–25% over 40 min at 8 mL/min. Fractions containing the desired product, identified by electrospray mass spectrometry (ESI-MS), were pooled, lyophilised, and stored at −20 °C.

KIIIA (CCNCSSKWCRDHSRCC*) with disulfide connectivity Cys1/Cys15, Cys2/Cys9, and Cys4/Cys16 was synthesised as previously described [23]. Briefly, peptide was assembled on Liberty Prime automatic synthesiser (CEM, Matthews, NC, USA) using rink amide-AM resin. Peptide was cleaved from solid support and side chains simultaneously deprotection in 92.5% TFA/2.5% TIPS/2.5% H_2_O/2.5% 3,6-dioxa-1,8-octanedithiol (DODT) for 2 h at room temperature. Excess TFA was evaporated by N_2_ flow, followed by peptide precipitation in ice-cold diethyl ether and centrifugation. Peptide was redissolved in 50% solvent B and lyophilised.

### 2.5. Oxidation

Reduced SxIIIC peptide (0.05 mg/mL) was oxidised at room temperature for 1 h in 0.1 M Tris-HCl pH 7.5, 0.1 mM ethylenediaminetetraacetic acid (EDTA), 1 mM glutathione oxidised, and 1 mM glutathione reduced. The reaction was quenched by adding formic acid to a final concentration of 5%. The final SxIIIC product was purified by RP-HPLC on a Gemini, 5 µm C18 110 Å, 250 × 10 mm column (Phenomenex, Torrance, CA, USA) using a linear gradient between 0–20% over 40 min at 3 mL/min. Fractions containing the desired product eluted at 10% solvent B and were pooled, lyophilised, and stored at −20 °C.

Reduced KIIIA peptide (0.3 mg/mL) was oxidised at room temperature for 24 h in 0.1 M NH_4_HCO_3_ pH 8.0, 0.81 mM glutathione oxidised, and 0.81 mM glutathione reduced. The KIIIA reaction was quenched with 1% TFA. The final KIIIA product was purified, as previously described [23].

### 2.6. Whole-Cell Patch-Clamp Electrophysiology

Automated whole-cell patch-clamp recordings were performed with a QPatch-16 automated electrophysiology platform (Sophion Bioscience, Ballerup, Denmark) using single hole (QPlate 16 with a standard resistance of 2 ± 0.4 MΩ) or multi-hole (QPlate 16X with a standard resistance 0.2 ± 0.04 MΩ–Na_V_1.8 only) planar chips on HEK293 cells heterologously expressing hNa_V_1.1–1.7/β1 (SB Drug Discovery, Glasgow, UK) or CHO cells heterologously expressing hNa_V_1.8/β3 in a tetracycline-inducible system (ChanTest, Cleveland, OH, USA). Recordings were acquired at 25 kHz and filtered with a Bessel filter at 8 kHz, and the linear leak was corrected by P/4 subtraction. Recordings were excluded from analysis if series resistance >10 MΩ.

Cells were maintained in Minimum Essential Medium Eagle (M5650) supplemented with 10% *v/v* fetal bovine serum, 2 mM L-glutamine, and the selection antibiotics geneticin, blasticidin and zeocin as recommended by the manufacturer. Cells were incubated at 37 °C with 5% CO_2_ and split every 3–4 days when reaching 70–80% confluency using the dissociation reagent TrypLE Express (Thermo Fisher Scientific, Scoresby, VIC, Australia). Cells were passaged 48 h prior to patch-clamp assay in a T-175 flask and cultured at 37 °C. Expression of hNa_V_1.8 was induced by the addition of tetracycline (1 μg/mL) for 48 h prior to assays. Cells were harvested at 80% confluence and dissociated with TrypLE Express and resuspended in Ham’s F-12 medium (Gibco) supplemented with 25 mM HEPES (Sigma Aldrich, St. Louis, MI, USA), 100 U/mL Penicillin-Streptomycin (Gibco) and 40 µg/mL trypsin inhibitor from *Glycine max* (soybean) (Sigma Aldrich), and allowed to recover with stirring for 30 min.

The extracellular solution consisted of (in mM) 145 NaCl, 4 KCl, 2 CaCl_2_, 1 MgCl_2_, 10 HEPES, and 10 glucose, pH to 7.4 with NaOH (adjusted to 305 mOsm/L with sucrose). The intracellular solution consisted of (in mM) 140 CsF, 1 EGTA, 5 CsOH, 10 HEPES, and 10 NaCl, pH to 7.3 with CsOH (adjusted to 320 mOsm/L with sucrose). SxIIIC was diluted in extracellular solution with 0.1% bovine serum albumin. Na_V_1.1–1.7 currents were elicited by a 50 ms pulse to −20 mV from a holding potential of −90 mV (repetition interval 20 s). Na_V_1.8 currents were elicited by a 50 ms pulse to +10 mV from a holding potential of −90 mV (repetition interval 20 s) in the presence of TTX (1 µM), which fully inhibits TTX-sensitive channels but not Na_V_1.8 at this concentration. Recordings were taken at ambient room temperature (22 °C) after 5 min incubation of each concentration. Peak current post-SxIIIC addition (I) was normalised to buffer control (I_0_). IC_50_s were determined by plotting difference in peak current (I/I_0_) and log peptide concentration. Concentration-response curves were fitted using the log (inhibitor) vs. response-variable slope (four parameters) Equation (1).
y = 1/(1 + 10^((LogIC_50_ − x) × HillSlope))(1)

Calculated IC_50_ were compared across subtypes and statistical differences determined by ordinary one-way ANOVA.

Current-voltage (I-V) curves were generated from a holding potential of −90 mV using a series of 500 ms step pulses ranging from −100 to +55 mV in 5 mV increments (repetition interval 5 s). Conductance-voltage (G-V) curves were generated by calculating the conductance (G) at each voltage (V) using the equation G = I/(V − Vrev), where Vrev is the reversal potential, and were fitted with a Boltzmann Equation (2).
y = 1/(1 + e^((V_50_ − x)/Slope))(2)

Voltage dependence of steady-state fast inactivation was assessed using a 10 ms pulse of −20 mV immediately after the 500 ms step (described above) to assess the availability of non-inactivated channels. Off rate data were acquired as previously described [22]. Briefly, using a holding potential of −90 mV and a 50 ms pulse to −20 mV every 20 s (0.05 Hz) peptides at IC_90_ concentration, SxIIIC (1 µM) and KIIIA (3 µM; determined in previous studies [23]), were incubated for 3 min before step-wise wash out with extracellular solution every 3 min for 35 min.

### 2.7. NMR Experiments

NMR experiments were carried out on a Bruker Avance III equipped with a cryoprobe (Bruker, Sydney, NSW, Australia). A sample of SxIIIC was dissolved in 90% H_2_O/10% D_2_O (*v*/*v*) at 1 mg/mL. All NMR experiments were conducted at pH 4.0, and one-dimensional (1D) ^1^H spectra, two-dimensional (2D) total correlated spectroscopy (TOCSY) and nuclear Overhauser effect spectroscopy (NOESY) experiments were run at 298 K. In addition, experiments including ^1^H–^15^N HSQC spectra and variable temperature 2D TOCSY (283–303 K) in 90% H_2_O/10% D_2_O (*v*/*v*) and ^1^H–^13^C HSQC in 100% D_2_O (*v*/*v*) at 298 K were also run.

### 2.8. NMR Structure Calculations

TopSpin 3.5 (Bruker, Sydney, NSW, Australia) was used to process all NMR spectra. Following the sequential assignment of SxIIIC using CCPNMR Analysis 2.4 [24], a list of inter-proton distances was generated. By comparing measured Hα chemical shifts to reported random coil Hα shifts [25], the secondary structure of SxIIIC was predicted. Hα, Cα, Cβ, HN chemical shifts derived from TOCSY, NOESY, ^1^H–^13^C HSQC, and ^1^H–^15^N HSQC spectra were used to calculate backbone (Φ and φ) dihedral angles constraints in TALOS-N [26]. Additional χ1 and χ2 side-chain dihedral angle restraints were derived using DISH [27]. Peak assignments were refined following several rounds using the ANNEAL function in CYANA v3.97 [28]. Additional H-bond restraints were included derived from temperature coefficient experiments in combination with D_2_O exchange data [29]. Finally, protocols in the RECOORD database were used in CNS to calculate 50 structures. These 50 structures were further refined in a water shell using CNS [30]. Based on lowest energy, fewest violations, and the best MolProbity scores [31], a final set of 20 structures were chosen to represent SxIIIC. The final 20 structure of SxIIIC have been deposited in the Protein Data Bank (PDB ID: 6X8R) as well as the Biological Magnetic Resonance Bank (BMRB ID: 30758).

## 3. Results

### 3.1. Activity-Guided Isolation of SxIIIC from Conus striolatus

Crude *C. striolatus* venom (~100 µg/mL) was fractioned using a linear gradient routinely used in-house for fractionation of cone snail venom (1% B/min). Given the limited material available, a single round of activity-guided fractionation using a high-throughput fluorescence assay assessing inhibition of veratridine-induced responses in the neuroblastoma cell line SH-SY5Y, expressing subtypes Na_V_1.2, 1.3, and 1.7 [32], identified compound(s) eluting in the buffer front on a Vydac 218TP C18 column as the bioactive components (Figure 1a). These were further purified by size-exclusion chromatography (Figure 1b), yielding a single bioactive fraction that was dominated by a mass of 2434.78 Da (Figure 1c). Sequence determination by Edman degradation identified this compound as a C-terminal amidated (*) novel µ-conotoxin with the sequence RGCCNGRGGCSSRWCRDHARCC*. The active peptide contained a cysteine framework consistent with the M-superfamily of conotoxins and therefore termed SxIIIC using standard µ-conotoxin nomenclature.

### 3.2. Peptide Synthesis of SxIIIC

Due to the limited amount of native material available, disulfide connectivity was not determined experimentally. Upon comparison of SxIIIC to the sequences of previously reported µ-conotoxins, SxIIIC was predicted to hold the canonical connectivity typically seen in the M-5 branch of the conotoxin M-superfamily of 1–4/2–5/3–6 [33]. Therefore, we produced synthetic SxIIIC with the disulfide connectivity of Cys3/Cys15, Cys4/Cys21, and Cys10/Cys22. SxIIIC was assembled using solid-phase peptide synthesis on rink amide-AM resin to produce C-terminal amidated peptide. Synthetic SxIIIC was thermodynamically oxidised and purified in high yield (80% based on pure linear starting material). Analytical RP-HPLC and ESI-MS were used to confirm crude linear and folded product (Figure 2a). While co-elution with native SxIIIC was not possible due to limited native material, the amide peaks of the 1D NMR spectrum (Figure 2b) are clearly distributed suggesting the peptide has a well-defined fold similar to previously described µ-conotoxins.

### 3.3. SxIIIC Displays a Unique Na_V_ Channel Subtype Selectivity Profile

We next evaluated the inhibition of synthetic SxIIIC of hNa_V_ channel subtypes by automated whole-cell patch-clamp electrophysiology in HEK293 cells expressing hNa_V_1.1–1.7 and CHO cells expressing hNa_V_1.8. SxIIIC potently inhibited hNa_V_1.4 (IC_50_ 15.11 ± 10.7 nM) and displayed ten-fold selectivity over subtypes hNa_V_1.1, 1.3, 1.6 and 1.7 (IC_50_ 132.0 ± 11.6, 89.4 ± 11.1, 124.9 ± 11.1 and 152.2 ± 21.8 nM, respectively) (Table 1 and Figure 3). Potent inhibition of hNa_V_1.4 is characteristic of most other µ-conotoxins. Unexpectedly, we observed a significant loss (<0.0001, one-way ANOVA) of activity against hNa_V_1.2 (IC_50_ 363.8 ± 53.8 nM) compared to hNa_V_1.4. SxIIIC did not affect the tetrodotoxin-resistant isoforms hNa_V_1.5 and hNa_V_1.8 when tested up to 1 μM, a trend consistent with other µ-conotoxins [34,35,36,37]. Together these results indicate that SxIIIC displays a unique µ-conotoxin selectivity profile of hNa_V_1.4 > hNa_V_1.3 > hNa_V_1.1 ≈ hNa_V_1.6 ≈ hNa_V_1.7 > hNa_V_1.2 >> hNa_V_1.5 ≈ hNa_V_1.8.

### 3.4. SxIIIC Is an Irreversible Presumptive Pore Blocker of Na_V_1.7

Additional experiments were conducted on hNa_V_1.7 to characterise the mechanism of action of SxIIIC. At hNa_V_1.7, SxIIIC (100 nM) elicited no change in channel voltage-dependence of fast inactivation (V_50_: control −63.6 ± 1.0 mV; SxIIIC −65.7 ± 0.9 mV) or activation (V_50_: control −22.7 ± 1.5 mV; SxIIIC −24.7 ± 1.9 mV), indicative of a pore blocker mechanism as observed for other µ-conotoxins (Figure 4a,b). Interestingly, SxIIIC (1 µM) irreversibly blocked current inhibition, as buffer washout over a 35 min time period was unable to recover any current (Figure 4c). In contrast, inhibition by the nearly irreversibly KIIIA (3 µM) could only be slowly reversed, with ~30% of current recovered over the washout period, consistent with previous reports [38]. Notably, under these conditions, TTX inhibition is rapidly reversible without evidence of accumulation of inactivated channels [23].

### 3.5. NMR Solution Structure of SxIIIC

2D TOCSY and NOESY NMR spectra were obtained and used for sequential assignment of individual amino acid spin systems. Secondary Hα chemical shifts were determined, and deviations from random coil Hα values were used to ascertain the presence of secondary structure within SxIIIC (Figure 5a) [25]. Consecutive negative deviations between Arg13 and His18 predicted an α-helix in this region. The presence of medium-range NOEs supports the observation of secondary structure (Figure 5b).

The three-dimensional (3D) NMR solution structure of SxIIIC was calculated using a simulated annealing protocol based on 173 NOE-derived distance restraints, and 12 Φ and 10 φ dihedral restraints derived from TALOS-N (Table 2) [26]. χ1 and χ2 angle restraints were derived by DISH for Cys3, Cys4, Cys15, Cys21 (χ1), Cys4, and Cys15 (χ2) [27]. Additional restraints included disulfide connectivities (Cys3/Cys15, Cys4/Cys21, and Cys10/Cys22) and hydrogen bonds determined through temperature coefficient and D_2_O exchange experiments (Arg16 → Ser12, Asp17 → Arg13, and His18 → Trp14). A total of 20 structures with no NOE violations (>0.2 Å) and no dihedral angle violations (>2 Å) were selected to represent based on the lowest total energy and MolProbity scores [31]. The final 20 structures were superimposed over backbone heavy atoms and the mean pairwise root-mean-square deviation (RMSD) was 0.76 ± 0.17 Å and 1.94 ± 0.39 Å for the all backbone and all heavy atom, respectively (Figure 5c). Energies and structural statistics for the NMR solution structure are shown in Table 2. The final structures were submitted to the Protein Data Bank (PDB ID: 6X8R) as well as the Biological Magnetic Resonance Bank (BMRB ID: 30758).

The NMR solution structure of SxIIIC is composed of a series of turns followed by a C-terminal α-helix between residues Arg13 and Asp17 (Figure 5d). Medium-range NOEs (Figure 5b) and hydrogen bonds (Arg16 → Ser12, Asp17 → Arg13, and His18 → Trp14) support the observation of the α-helix. SxIIIC is stabilised by three disulfide bonds with Cys3/Cys15 anchoring the N-terminal to the α-helix.

## 4. Discussion

Na_V_1.7, along with Na_V_1.1, 1.3, 1.6, 1.8, and 1.9, plays an important role in pain signalling and has shown promise as a drug target for pain therapies. Toxins isolated from highly venomous creatures have evolved to potently and specifically target Na_V_ channels and offer a great starting point as drug leads. However, a major limitation in their success has been the high sequence homology of the Na_V_ channel pore, which could lead to undesirable side effects caused by off-target inhibition. As most toxins modulators that target the therapeutically relevant subtype Na_V_1.7 act as gating modifiers and bind to the voltage-sensing domain, they offer limited insights into the channel pore interactions. Despite µ-conotoxins Na_V_1.7 pore blockers being less potent than gating modifiers, they may be used to further our understanding of potency and selectivity for Na_V_ channel. In this body of work, we have isolated a novel µ-conotoxin, SxIIIC, from *C. striolatus*, characterised the selectivity profile against hNa_V_ channel subtypes and determined the NMR solution structure.

We first used activity-guided fractionation to isolate native SxIIIC from crude venom and subsequently determined the peptide sequence by Edman degradation. This sequence was identical to STRIO_41, recently reported from a *C. striolatus* transcriptome [40]. The unusual initial hydrophilic elution profile of native SxIIIC may be explained by the preparation of crude venom in 30% acetonitrile which artificially attributed a higher gradient and may account for lack of identification in earlier proteomic studies [40]. When we synthetically produced SxIIIC the peptide eluted at ~10% acetonitrile on a Gemini, 5 µm C18 column. We next evaluated the pharmacological inhibition of hNa_V_ channel subtypes by automated whole-cell patch-clamp electrophysiology. SxIIIC displayed potent inhibition of hNa_V_1.4 (IC_50_ 15.11 ± 10.7 nM) which is characteristic of most other µ-conotoxins, and may provide an evolutionary advantage as Na_V_1.4 is expressed exclusively in skeletal muscle and cone snail venom acts rapidly to immobilise prey [41,42]. Interestingly, when compared to hNa_V_1.4, synthetic SxIIIC was significantly less potent against hNa_V_1.2 (IC_50_ 363.8 ± 53.8 nM). This was surprising as many other µ-conotoxins preferentially target Na_V_1.4 and Na_V_1.2 [37], with the exception of BuIIIB which preferences Na_V_1.3, as mentioned earlier [18]. However, many previous studies used rat and mouse isoforms as opposed to human, and species specificity may explain this variation. Another explanation may be the disulfide connectivity. Due to limited quantity of native material, we were unable to unambiguously determine disulfide connectivity. However, when we produced synthetic SxIIIC we observed a single major peak following thermodynamic oxidation of synthetic SxIIIC. Moreover, the biological activity of synthetic peptide at Na_V_1.7 is consistent with that of the native material, which inhibited Na_V_1.7 expressed in SH-SY5Y cells. Biological activity of the alternate folding arrangements remains to be determined.

Although SxIIIC was not selective for hNa_V_1.7, potent inhibition of this isoform is nonetheless unusual for the µ-conotoxins. Only two µ-conotoxins have reported nanomolar activity against hNa_V_1.7 isoforms: KIIIA and CnIIIC (485 nM) [20]. In 2011, McArthur et al. showed KIIIA inhibited hNa_V_1.7 with an IC_50_ of 147 nM [19]. However, as these channel isoforms lacked β subunits it is difficult to directly compare results. In our hands, in a comparable study by Tran et al. [23], KIIIA with native connectivity exhibited an IC_50_ of 363 nM which is approximately two-fold less active than SxIIIC. At hNa_V_1.7, the wash-off rate of SxIIIC was markedly slower than that of KIIIA assessed under identical experimental conditions. Notably, previous reports of a more readily reversible KIIIA [19] could be due to the step-wise, rather than continuous, solution exchange necessitated by using automated patch-clamp, and the relative lack of control of seal quality in these systems. In addition, the aforementioned co-expression with the β1 subunit, as well as use of KIIIA isomer 1 with disulfide connectivity (Cys1/Cys15, Cys2/Cys9, and Cys4/Cys16) may contribute to these seemingly disparate results. However, given that TTX is readily reversible under the same conditions [23], and we do not observe any accumulation of inactivated channels with our pulse protocol, it is likely that a comparably slower off-rate of SxIIIC contributes to its greater potency at hNa_V_1.7. Notably, the protocols used to compute IC_50_ values at each subtype do not directly permit calculation of on-rates, although these appeared comparatively slower at Na_V_1.4 and Na_V_1.7 (Appendix A). Furthermore, the relatively slow on-rate of SxIIIC resulted in steady-state inhibition only being achieved at higher concentrations. While this may be circumvented by substantially longer incubation times, this is difficult to achieve experimentally. As such, it is possible that our calculations are an under-estimate of the true IC_50_ values.

Given the high sequence homology between µ-conotoxins (Figure 1d) and the highly conserved nature of Na_V_ channel subtypes, particularly within the pore region, these findings may indicate subtle differences that could be exploited to further our understanding of potent Na_V_1.7 inhibition. Compared to KIIIA, the N-terminal tail of SxIIIC is extended by two residues, while the peptide sequence between the second and third cysteines (referred to as loop 1) is extended by four residues. It is possible that the extended loops of SxIIIC permit more interactions with Na_V_ channels than KIIIA, and we thus sought to determine the NMR structure of SxIIIC by 2D homonuclear NMR for comparison to other μ-conotoxins and to provide a foundation for future structure-activity relationship studies. SxIIIC structure is composed of a C-terminal α-helix and is stabilised by three disulfide bonds. A marginally shorter α-helix seen in the final structure compared to the earlier predicted structure is likely explained by additional restraints applied during refinement.

Recent landmark cryo-EM studies of hNa_V_ channels provide insights into the molecular basis of µ-conotoxin binding [14,43]. The calculated SxIIIC structure shared the characteristic α-helix of other M-5 branch µ-conotoxins KIIIA, BuIIIB, and SIIIA [18,35,38]. Residues extending from this helix have been shown to play an important role in Na_V_ channel inhibition, including Lys7, Trp8, Arg10, Asp11, and His12 (KIIIA numbering) [38,44,45]. SxIIIC is almost identical to KIIIA in this region, with the exception of Arg13 (SxIIIC) which has a conservative substitution from Lys7 in KIIIA. However, this residue is either an Arg or Lys at this position in all other µ-conotoxins and it is unsurprising that the cryo-EM structure has shown Lys7 in KIIIA to play an important role in channel binding, with the side chain extending down into the pore and acting to occlude the outer channel mouth [14]. It is likely that the analogous residue in SxIIIC similarly contributes to activity, although this remains to be experimentally confirmed. Compared to KIIIA, both the N-terminus and loop 1 of SxIIIC are extended. When SxIIIC is superimposed with the KIIIA/Na_V_1.2 structure (Appendix A), residues in loop 1 of SxIIIC appear to clash with the extracellular pore loop of domain I. As these are dynamic environments, it is likely that SxIIIC shifts from this proposed binding mode to a more energetically favourable position, and in doing so may contribute additional contacts to the channel that could contribute to its irreversible inhibition and increase in potency at Na_V_1.7 compared to KIIIA. Furthermore, the N-terminal extension has been shown to influence µ-conotoxin selectivity for neuronal (Na_V_1.2) and skeletal subtypes (Na_V_1.4) and may have influenced the observation of decreased potency at Na_V_1.2 [46]. Interestingly, µ-conotoxin SmIIIA, which shares 90% sequence identity with SxIIIC (Figure 1d), has an extended N-terminal (Pyroglutamate1 and Gly2) and an additional charged residue in loop 1 compared to SxIIIC, but only weakly inhibits mammalian Na_V_1.7 (IC_50_ 1.3 ± 0.23 µM) [37]. Further investigation of the structure-activity relationships of these peptides may help us understand the role of these extensions in Na_V_ channel inhibition and selectivity.

In conclusion, we discovered a novel µ-conotoxin SxIIIC, produced the peptide synthetically and evaluated its pharmacological and structure characteristics. To date, SxIIIC is one of the most potent µ-conotoxin hNa_V_1.7 inhibitors, acting as a presumptive pore blockers, and presents as a promising peptide for Na_V_ channel inhibitor development.

## Figures and Tables

**Figure 1 biomedicines-08-00391-f001:**
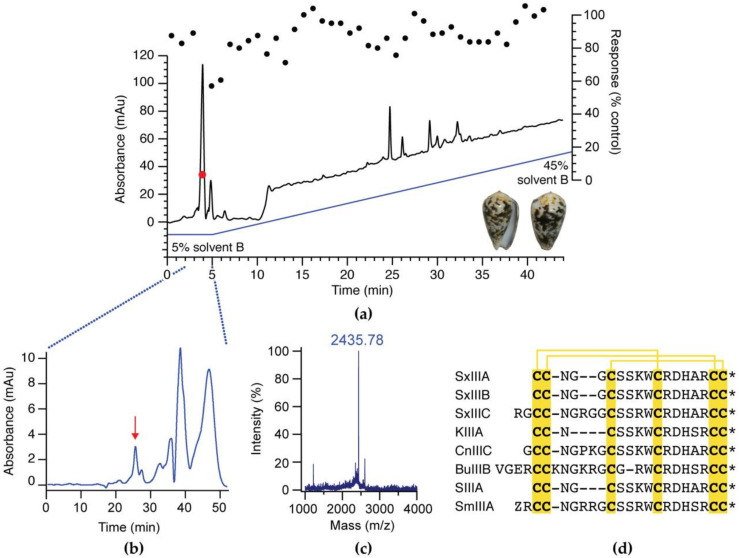
Activity-guided isolation of SxIIIC from *C. striolatus* venom. (**a**) Crude venom (200 µg) isolated from one specimen of *C. striolatus* (inset) was fractionated and assayed for Na_V_ inhibitor activity by fluorescence imaging in SH-SY5Y cells expressing subtypes Na_V_1.2, 1.3, and 1.7 (right *y*-axis; the activity of each fraction represented by circles overlaying the chromatogram). The bioactive component (red circle) eluted early. (**b**) Size-exclusion chromatography was used to further purify fractions collected between 2 and 5 min to isolate the bioactive component. The active fraction (red arrow) was dominated by (**c**) a mass of 2435.78 Da (m/z) and subjected to sequencing by Edman degradation, which identified a novel µ-conotoxin with a C-terminal amidation named SxIIIC. (**d**) Sequence alignment of SxIIIC with previously reported µ-conotoxins. * denotes C-terminal amidation.

**Figure 2 biomedicines-08-00391-f002:**
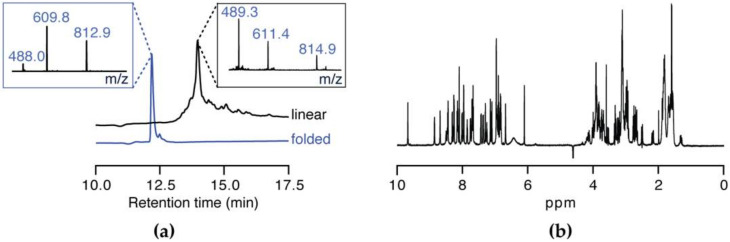
Peptide synthesis of SxIIIC. (**a**) RP-HPLC trace and ESI-MS spectra (insert) after cleavage (top) and oxidation (bottom) confirm successful synthesis and purification of SxIIIC. (**b**) 1D ^1^H NMR spectrum of SxIIIC displays well-distributed amide peaks indicating a peptide with a definitive fold.

**Figure 3 biomedicines-08-00391-f003:**
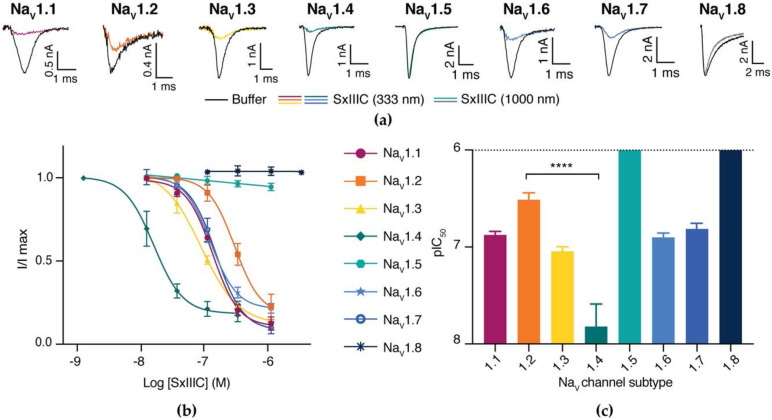
Activity of SxIIIC across Na_V_ channel subtypes assessed by whole-cell patch-clamp electrophysiology. (**a**) Representative hNa_V_1.1–1.8 current traces before (black line) and after addition of SxIIIC (333 or 1000 nM; coloured line). Currents were obtained by a 50 ms pulse to −20 mV for hNa_V_1.1–1.7 and +10 mV for hNa_V_1.8. (**b**) Concentration-response curves and (**c**) comparative IC_50_ potency of SxIIIC at hNa_V_1.1–1.8. SxIIIC most potently inhibited Na_V_1.4, with 10-fold selectivity over hNa_V_1.1, 1.3, 1.6 and 1.7, 24-fold selectivity over hNa_V_1.2 (****, *p*-value < 0.0001) and >100-fold selectivity over hNa_V_1.5 and hNa_V_1.8. Data are presented as mean ± SEM, with *n* = 3–10 cells per data point, see Table 1.

**Figure 4 biomedicines-08-00391-f004:**
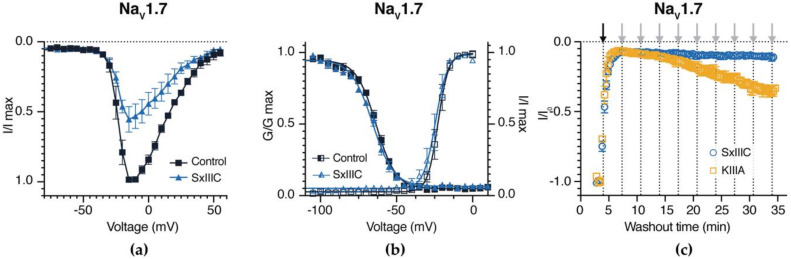
Effect of SxIIIC on the electrophysiological parameters of hNa_V_1.7. (**a**) Voltage-current relationship curves before (black squares) and after addition of SxIIIC (blue triangles). SxIIIC (100 nM) inhibited peak current but did not affect the membrane voltage of peak current. (**b**) Conductance-voltage relationship before (black open squares) and after addition of SxIIIC (blue open triangles) and voltage-dependence of steady-state fast activation curves before (black squares), and after addition of SxIIIC (blue open triangles). SxIIIC (100 nM) did not shift the V_1/2_ of voltage-dependence of activation or V_1/2_ of steady-state fast inactivation at hNa_V_1.7. (**c**) Wash out after addition of equipotent peptide (IC_90_), SxIIIC (1 µM; blue circles) and KIIIA (3 µM; yellow squares). Black arrow indicates peptide addition and grey arrows indicate each subsequent wash out step. Current inhibition was irreversible for SxIIIC over 35 min and was almost irreversible for KIIIA. Data presented as mean ± SEM, with *n* = 4 cells per data point.

**Figure 5 biomedicines-08-00391-f005:**
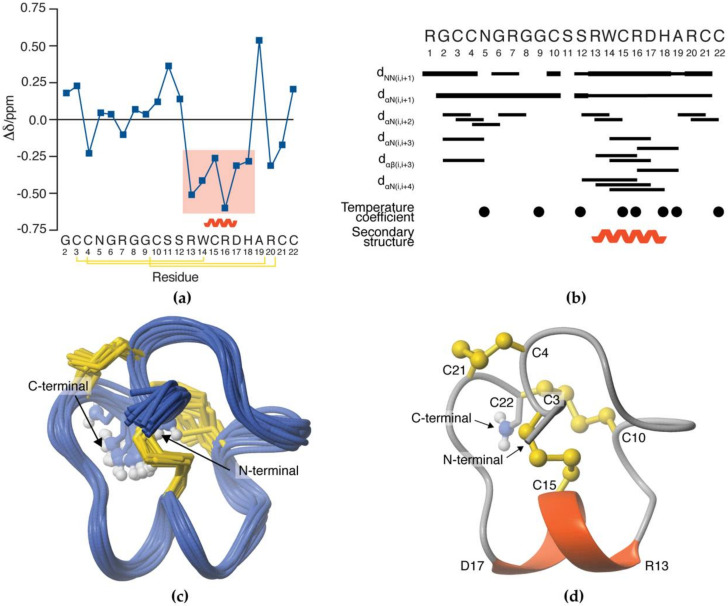
3D NMR solution structure of SxIIIC. (**a**) Secondary Hα chemical shift deviations from random coil values predicted the 3D structure to include an α-helix between Arg13–His18 (shaded box). Disulfide connectivity indicated by yellow lines. (**b**) Summary of local and medium-range NMR data of SxIIIC. Bar width represents the strength of the NOE. Closed circles indicate residues involved in solvent protection as determined by temperature coefficient experiments. The α-helix region is denoted by a curved line. (**c**) The 20 best conformations of NMR solution structure as analysed by MOLMOL superimposed across Cys3 and Cys22 [39]. (**d**) The final 3D structure of SxIIIC following CNS refinement in a watershell is composed of an α-helix between residues Arg13–Asp17 (red helix) constrained by three disulfide bonds (yellow).

**Table 1 biomedicines-08-00391-t001:** Potency of SxIIIC across Na_V_ channel subtypes.

Subtype	IC_50_ ± SEM (nM)	*n*
Na_V_1.1	132.0 ± 11.6	5
Na_V_1.2	363.8 ± 53.8	5
Na_V_1.3	89.4 ± 11.1	3
Na_V_1.4	15.11 ± 10.7	10
Na_V_1.5	>5000	4
Na_V_1.6	124.9 ± 11.1	4
Na_V_1.7	152.2 ± 21.8	4
Na_V_1.8	>5000	3

**Table 2 biomedicines-08-00391-t002:** Statistical analysis of SxIIIC NMR solution structure.

**Distance Restraints**	
Intraresidue (i − j = 0)	77
Sequential (|i − j| = 1)	58
Medium range (|i − j| < 5)	24
Long range (|i − j| > 5)	7
Hydrogen bonds ^1^	6
Total	172
Dihedral angle restraints	
Φ	12
Φ	10
χ1	4
χ2	2
Total	28
**Structure Statistics**	
Energies (kcal/mol, mean ± SD)	
Overall	−465.4 ± 17.8
Bonds	8.8 ± 0.9
Angles	33.4 ± 2.5
Improper	16.2 ± 2.8
Dihedral	96.6 ± 1.3
Van de Waals	−73.1 ± 6.3
Electrostatic	−547.7 ± 19.5
NOE (exp.)	0.1 ± 0.0
Constrained dihedrals (exp.)	0.1 ± 0.1
Atomic RMSD (Å)	
Mean global backbone (1–22)	0.76 ± 0.17
Mean global heavy (1–22)	1.94 ± 0.39
**MolProbity Statistics**	
Clash score, all atoms ^2^	14.8 ± 6.6
Poor rotamers	0.0 ± 0.0
Ramachandran outliers (%)	0.0 ± 0.0
Ramachandran favoured (%)	80.0 ± 10.0
MolProbity score	2.4 ± 0.3
MolProbity percentile ^3^	54.0 ± 13.6
**Violations**	
Distance constraints (>0.2 Å)	0
Dihedral-angle constraints (>2°)	0

^1^ Two hydrogen bond restraints included per bond. ^2^ Number of steric overlaps (>0.4 Å)/1000 atoms. ^3^ 100% is the best among structures of comparable resolution. 0% is the worst.

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
