# Peer review of "Discovery, Pharmacological Characterisation and NMR Structure of the Novel µ-Conotoxin SxIIIC, a Potent and Irreversible NaV Channel Inhibitor"

_biomedicines, 2020, doi:10.3390/biomedicines8100391_

Round 1
Reviewer 1 Report
This article reports the isolation and characterization of a novel peptide SxIIIC from the Conus striolatus, its effects on hNav channel subtypes by automated whole-cell patch-clamp electrophysiology in HEK293 and CHO cells expressing these channels. As a result, authors found that SxIIIC is the most potent human Nav1.7 pore blocker to date with unique Nav channel selectivity profile. It is concluded that Nav channel blockers may be useful as molecular tools for chronic pain drug development. These results are an excellent addition to the investigation of µ-conotoxins inhibiting Nav1.7 channels.
I have a few small suggested edits and additions to this manuscript that I think will improve clarity of the methods and context of this study. I listed these revisions below.
- Please add to experimental 2.3 section, line 96, description and figure of last stage of SxIIIC isolation, RP-HPLC. This information appear only in Discussion section, line 352. Line 100, please write the name of company and country where the column was made. Line 100, please explain what does this mean “into 45 x 1 min fractions”. Line 102, what wavelength for detection was used? Line 103, please explain what does this mean “orthogonal fractionation”. Section 2.4, please delete information about HPLC.
- Line 244 and Figure 1C, peptide mass 2435.7832, too high accuracy for MALDI. If you used also ESI-MS for identification synthetic peptide, please add ESI-MS to the section Mass spectrometry and move it after section 2.6.
- Line 317 and Figure 5D. Please check, it should be “Asp17” not “Arg17”.
- Line 359. It would be better to add information about BuIIIB to the Introduction.
- Line 377. Please check, “between the second and third cysteines (referred to as loop 1) is extended by three residues”. “Four”?
- Lines 395-396. “When SxIIIC is superimposed with the KIIIA/NaV1.2 structure, residues in loop 1 of SxIIIC appear to clash with the extracellular pore loop of domain I”. Is it your experiment, I do not see this in methods? If it is not, please provide some support for this statement.
- Lines 346-347. Please rephrase this sentence “We first used activity-guided fractionation to isolate native SxIIIC from crude venom and Edman degradation to determine peptide sequence.”
- Lines 350-351. Please add reference after the statement.
- Line 89. Not “foetal” but “fetal”.
- Line 115. “at 56 °C”, do you use such methodology or it is a mistake?
- Line 223. Not “SxIIIC” but “SxIIIC”.
- Line 232. “Analytical RP-HPLC and ESI-MS were used”.
- Line 234. Not “1D NMR spectra” but “1D NMR spectrum”. Line 250 the same.
- Line 236, Figure 1A. Why the baseline goes up and not straight? What is the purity grade of the acetonitrile?
- Line 264, Figure 3B. Please add “Nav” before of subtype designations of the channels.
- Line 300. Figure 5 comes after the table 2. The meaning should be the other way around
- Line 304. “and 12 Φ and ten φ dihedral”. Please change the font.
- Line 306. “Cys4, Cys15, and Cys21 (χ1), and Cys4, and Cys15 (χ2)”.
- Line 310. “the lowest total energy, and MolProbity score [30], and”.
- Line 311. “submitted to the Protein Data Bank (PDB ID: 6X8R) and as well as Biological Magnetic Resonance Bank”.
- Line “Nav1.7 plays”.
- Line 351. ”SxIIIC with the canonical disulfide connectivity”.
- Line 352. “peptide was eluted”.
- Line 369. Please, add reference [18, 19].
- Line 443. “Pharmacol. Ther.”, Line 466. “Br. J. Pharmacol.”,Line 517. “Conus striolatus”.
- The comma before "and" in the lines: 34, 38, 43, 66, 134, 135, 144, 149, 151, 208, 217, 229, 240, 292, 306, 307, 308.310, 318, 335, 386, 388. I have found comma before "and" lines 167, 169, and in other cases is missing.
Reviewer 2 Report
The article "Discovery, pharmacological characterisation and NMR structure of the novel μ-conotoxin SxIIIC, a potent and irreversible Nav channel inhibitor" by McMahon et al. describes a novel Nav channel inhibiting conotoxin from Conus striolatus bearing the classical µ-conotoxin fold. The work is cleanly performed and succinctly presented however some of the assertions presented require further substantiation.
Major concerns
- SxIIIC is reported to display a “unique” Nav channel selectivity profile (1.4 > 1.3 > 1.1 » 1.6 » 1.7 > 1.2 >> 1.5 » 1.8) in particular regard to lower Nav1.2 potency. However, the representative Nav1.2 (and Nav1.1) current traces shown in the presence of SxIIIC in Fig.3 display very unusual kinetics suggestive of either a. Insufficient clamp control, or b. (an even more unexpected) modification of gating by the peptide. If the first option would be the culprit it is likely that the IC50 estimated for Nav1.2 is not reliable therefore it is advised to verify the data from those experiments and reassess the IC50. If gating modification is indeed occurring, it surely merits further research.
- Admittedly, is very likely that SxIIIC functions as a Nav pore blocker, however it cannot be ascertained without experimental verification. Please use "putative" or "presumptive pore blocker" where needed or provide experimental confirmation of this MoA.
- Like other “mini” μ-conotoxins, KIIIA and CnIIIA, SxIIIC inhibits human Nav1.7 but it does so in an irreversible fashion. However, the Nav1.7/KIIIA washout behaviour reported does not recapitulate McAthur et al 2011 report (that shows full KIIIA washout after ~35mins). This is important as KIIIA is used as an established control for reversibility. This raises question concerning the extent of reversibility attainable under the experimental conditions used in the current report. Experimental difference between automated and manual patch-clamp, partial solution replacement and full bath perfusion and 50ms vs 10ms stimulation pulses may underlie the disagreement. To that regard please consider that: a. the seal quality in manual patch-clamp outperforms that of automated platforms, please include a sentence stating this limitation when comparing to manual patch data; b. the fluidic setup of automated patch rigs like the Qpatch does not warrant full solution exchange, this can be easily verified by washing out TTX (at IC90) to determine the speed and which the solution is fully exchanged; and c. excessively long stimulation pulses drive mammalian Navs into deeper non-conductive inactivated states that can be confounded with irreversibly blocked non-conductive states and/or underscore state dependence. To address that 1.- perform the "washout" experiment at IC50 for both peptides (50ms pulses). 2.- perform the experiments at IC50 and IC90 with 10ms pulses.
Minor concerns
- Provide equations used for IC50 calculations and Boltzmann fit.
- Provide details on frequency of stimulation during peptide washin, IV and SSI.
- Provide average diary plots or representative examples for washin/washout in all channel isoforms studied.
- Provide a table including kon, koff and KD for all channel isoforms.
- Provide activation and inactivation plots and fit parameters for all isoforms in control and toxin.
- Provide exact n value (not ranges) for Nav IC50 determination.
- It is impressive to see ~4nA of Nav1.8 current in heterologous systems let alone in CHO cells. Are the Nav stable cell lines used for APC correctly identified?
Round 2
Reviewer 2 Report
- Define all parameters, including “Bottom” and “Top”, on Equations (1) and (2).
- Lines 398-399: “…However, as steady-state inhibition is achieved rapidly at higher concentrations, it is clear that SxIIIC likely has higher affinity (lower Kd) for NaV1.7 than NaV1.2…” The representative washins for Nav1.2 and Nav1.7 provided in S2 overlap substantially. Given that those are the only examples shown and no attempt at quantification of binding kinetics was provided it is not at all clear that SxIIIC has higher affinity (lower Kd) for Nav1.7 than Nav1.2. Please rephrase the sentence (lines 398/399) to reflect the uncertainty of the aforementioned statement or provide the relevant quantification.
- For Nav1.7 and Nav1.4 steady-state inhibition is only achieved at higher concentrations under the presented experimental regime. Please briefly discuss the implications.
- State the concentration of TTX used during the Nav1.8 recordings. Note that TTX can interact with the Nav1.8 pore albeit at comparatively high [TTX]. For instance, at 1uM TTX, ~20% of Nav1.8 channels would be occupied by TTX. Furthermore, KIIIA loses ~20-fold potency against TTX occupied Nav1.2 (PMID: 20410356). To assess potency of a pore blocker in the presence of another requires careful consideration and controls.
